

# Transition to conservation agriculture: how tillage intensity and covering affect soil physical parameters

Felice Sartori[1], Ilaria Piccoli[1], Riccardo Polese[1], Antonio Berti[1]

[1] DAFNAE Department, University of Padova, Viale Dell'Università 16, 35020, Legnaro, PD, Italy

*Correspondence to*: Ilaria Piccoli (ilaria.piccoli@unipd.it)

**Abstract.** Conservation agriculture (CA) relies on two key practices to improve agricultural sustainability—reduced tillage and cover crop usage. Despite known soil physics benefits (reduced soil compaction and strength, enhanced soil porosity and permeability), inconsistent reports on short-term CA results have limited its adoption in European agroecosystems.

To elucidate the short-term effects, a three-year experiment in the low-lying Venetian plain (Northern Italy) was undertaken.

Bulk density, penetration resistance, and soil hydraulic measures were used to evaluate results obtained by combining three tillage intensities (conventional tillage (CT), minimum tillage (MT), no tillage (NT)) with three winter soil coverages (bare soil (BS), tillage radish cover crop (TR), winter wheat cover crop (WW)).

Among the tillage methods and soil layers, CT, on average, reduced BD (1.42 g cm$^{-3}$) and PR (1.64 MPa) better in the 0-30 cm tilled layer. Other treatments yielded higher values (+4% BD and +3.1% PR) in the same layer. Across the soil profile,

reduced tillage coupled with WW improved soil physics even below the tilled layer, as evidenced by root growth-limiting threshold declines (-11% in BD values >1.55 g cm$^{-3}$ and -7% in PR values >2.5 MPa). Soil hydraulic measures confirmed this positive behaviour; NT combined with either BS or WW produced a soil saturated conductivity of $2.12 \times 10^{-4}$ m s$^{-1}$ (four-fold that of all other treatments). Likewise, sorptivity increased in NT combined with BS *versus* other treatments (3.64 $\times 10^{-4}$ m s$^{-1}$ *vs* an all-treatment average of $7.98 \times 10^{-5}$ m s$^{-1}$). Our results suggest that despite some measure declines due to

reduced tillage, the strategy enhances soil physics. In the short term, cover crop WW moderately increased physical soil parameters, whereas TR had negligible effects. This study demonstrates that CA effects require monitoring several soil physical parameters.

## 1 Introduction

Minimal soil disturbance, permanent soil covering, and crop rotation represent the main pillars of Conservation Agriculture

(CA) (FAO, 2017). Adoption of CA not only leads to reduced labour and farm costs, but also provides several ecosystem services that increase agroecosystem sustainability. Its hallmarks of reduced soil tillage, applied cover crops (CC), and rotated crops generally improve the physical parameters of soil and foster nutrient cycling and soil biological activity. In general, CA has been shown to enhance most soil physical properties, but some contrasting results have been reported (Blanco-Canqui and Ruis, 2018). Negative outcomes have often been obtained in no tillage (NT) systems that failed to



specify whether or not the soil was permanently covered between two main crops. Typically, CCs are used to maintain soil coverage. It consists of cultivating plants between two main crops, leaving the entire biomass on the field after the growing season, and eventually burying it before the subsequent crop is planted (Schipanski et al., 2014). The use of CC is a pivotal strategy for enhancing soil physical properties in reduced tillage systems (Blanco-Canqui et al., 2011).

Despite a growing interest in CA from many agroecosystems and especially in the Americas, European adoption of the
practice has faltered (Kassam et al., 2019). One reason behind limited CA adoption in Europe is uncertainty about its effects during the transitional period after conversion from conventional to conservation agriculture (Pittelkow et al., 2015; Rusinamhodzi et al., 2011). Site-specific trials offer not only a chance to expand what is known about the impact of CA on soil physical parameters, but also an opportunity to determine an optimal tillage—CC combination capable of mitigating local soil threats while simultaneously reducing conversion-time side effects. Indeed, under specific conditions, occasional
tillage is recommended (Liu et al., 2016), whereas in other situations, implementation of minimal tillage (MT) may provide benefits equal to those of NT (Chen et al., 2017; Teodor et al., 2009). Moreover, efficient use of CC requires careful selection of species, seeding date, and management strategy (Daryanto et al., 2018). Differing species may positively impact nutrient cycling, soil properties, and/or weed suppression, although such factors must be cost-effective, since they do not contribute directly to profitability (Ranaldo et al., 2019; Schappert et al., 2019).

In the low-lying Venetian Plain of Northern Italy, soils contain low organic carbon, high carbonate, and are micro-structured. The principal threats to such soils are organic matter depletion and compaction (Piccoli et al., 2020). Traditionally, farmers have countered compaction with annual deep ploughings that, in the long-term, may contribute to plough pan formation and foster organic matter mineralization. Among the benefits of CA is its potential to improve soil structure along the full soil profile, while protecting soil organic matter (Hobbs, 2007; Thomas et al., 1996). Nonetheless, contrasting results have been
reported, especially in the early years after CA adoption. In general, negative reports of the short-term effects of CA on physical soil parameters seem limited to bulk density (Guan et al., 2014), soil strength (Munkholm et al., 2003; Palm et al., 2014), and soil saturated hydraulic conductivity (Buczko et al., 2006). The use of CC to minimise the side effects of NT or MT represents a valuable short-term solution to facilitate conversion from conventional agriculture to CA. If cash crops are grown during the spring and summer, then autumn-drilled CC must develop rapidly to cover the soil before winter, and
devitalisation must occur in the spring before cash crop seeding.

A suitable CC species for northern Italy agroecosystems is *Poaceae* (e.g., wheat, barley, oat, ray, and triticale), which already is well adapted and easily managed by farmers. *Poaceae* can control weeds and reduce nutrient losses. Moreover, its fibrous root apparatus can positively impact soil physical properties, especially in the shallow soil layer (García-González et al., 2018). Alternatively, to mitigate soil compaction and improve the physical quality of the soil, tillage radish (*Raphanus*
*sativus L.*) "TR" has been broadly applied as a CC (Ciaccia et al., 2019; Crotty and Stoate, 2019). TR is a brassicaceous plant, specifically selected to improve the macro-porosity and pore connection of soil. Its 5 cm (D) × 30 cm (L) taproot counters soil compaction while enhancing water infiltration. While it is killed in the winter, it is easily managed in the spring (in a NT system also) (Büchi et al., 2020). As has been demonstrated by the limited use of CC throughout northern Italy,



there is a general lack of knowledge on TR adaptability in such agroecosystems, and its effectiveness at improving soil
properties.

The evolution of soil physical traits is frequently done by measuring soil Bulk Density (BD), soil Penetration Resistance (PR) and soil infiltration (Blanco-Canqui and Ruis, 2020). These indicators of soil strength, soil porosity and water and gas permeability are evaluated from measures using different scales. Specifically, PR is evaluated most often using penetrometers having probes of a few centimetres in diameter, BD is determined from undisturbed soil core samples having
a slightly larger diameter, and soil infiltration measures typically rely on infiltrometers of a far larger size (Al-Shammary et al., 2018; Dexter et al., 2007; Morbidelli et al., 2017). These different measurement scales can greatly affect results, particularly in no-till soils, where not only root penetration, but water and gas penetration, can be principally affected by the presence of bio-pores that create preferential pathways for root development even in what seem like highly-compacted soils. The goals of this study are to evaluate soil physical traits using these different spatial resolution measures during the
transition from conventional tillage to CA. For this purpose, BD, PR, and soil hydraulic parameters were monitored from 2018 to 2020 in field surveys conducted on trials created by combining three different tillage system with three winter soil coverings.

## 2 Materials and methods

The experiment took place at the Lucio Toniolo Experimental Farm, located in Legnaro, PD (NE Italy, 45° 21 N; 11° 58 E; 6
m a.s.l.), where the climate is sub-humid, with temperatures between -1.5°C on average in January and 27.2°C on average in July. Rainfalls reach 850 mm annually, with a reference evapotranspiration of 945 mm that exceeds rainfalls during April to September. Highest rainfalls occur in June (100 mm) and in October (90 mm), while winter is the driest season with average rainfalls of 55 mm. The shallow water table ranges from 0.5 to 2 m in depth, with the lowest values recorded in summer.

The trial, begun in spring 2018, was designed as a split plot, with two replicates. A 2-ha area was divided into 18 plots of
about 1.111 m$^2$ each. Soil at the site is Fluvi-Calcaric Cambisol (FAO-UNESCO, 2008) with a silty loam texture.

At the start of the experiment, the average soil texture of each plot was determined by laser diffraction (Malvern Mastersizer 2000; Malvern Instruments, Malvern, UK) as described in Bittelli *et al.*, 2018. Three different tillage treatments were randomized in plots: the conventional tillage (CT) plot was ploughed to 30 cm and harrowed (15 cm), the minimum tillage (MT) plot was tilled to a depth of 15 cm and then harrowed, and the no tillage (NT) plot was sod-seeded. Then, three winter
soil coverings were randomized within each of these plots: TR (*Raphanus sativus L.*), winter wheat (WW – *Triticum aestivum L.*), and bare soil (BS), where no soil cover was present other than the residues from the crop of the previous year. Cover crops were drilled on the main crop residues in autumn 2018 and 2019. The main crop was always maize (*Zea mays L.*).



## 2.1 Field surveys

Four parameters were selected to monitor soil physical qualities: bulk density (BD), penetration resistance (PR), and saturated hydraulic conductivity (Ks) together with sorptivity (S). The survey timetable is shown in Figure 1.

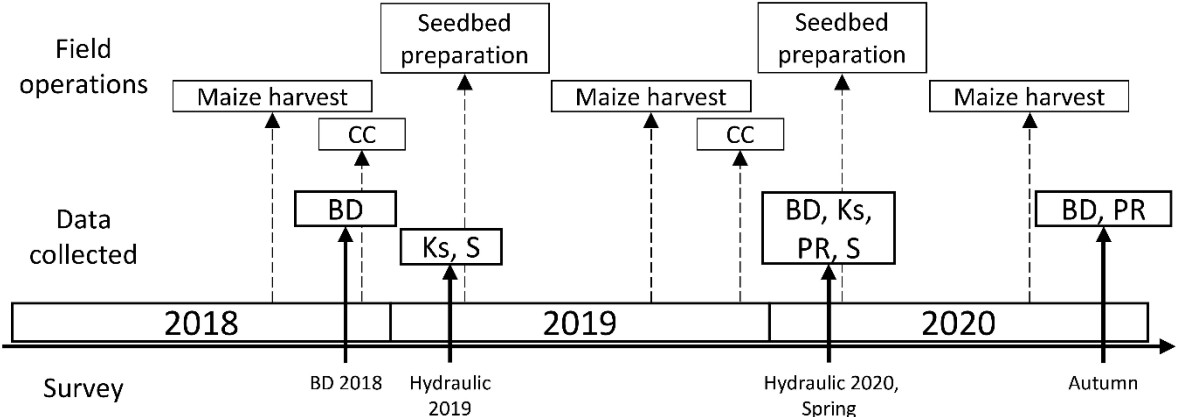

**Figure 1. Survey timetable. BD: bulk density, CC: cover crop seeding, Ks: saturated hydraulic conductivity, PR: penetration resistance, S: sorptivity**

## 2.1.1 Bulk density


The surveys were conducted on three sampling dates. Measurements were first performed at the start of the experiment after the first-year harvest (BD 2018, time 0). The second collection occurred in 2020 before tillage operations and after CC devitalization. The final sampling was performed in the same year, after the maize harvest but prior to soil preparation and subsequent crop seeding. Hereafter, the first, second, and third BD surveys will be referred to as "2018", "Spring", and 105 "Autumn", respectively. Each soil core was considered in 10 cm layers, which yielded six different depth-linked BD values from each sample. All samples were oven dried (24 hr at 105°C) to calculate BD (core method) (Grossman and Reinsch, 2002) on undisturbed 7 cm diameter soil cores that were collected with a hydraulic probe from the 0-60 cm layer.

### 2.1.2 Penetration resistance

Penetration Resistance (PR) was measured with a penetrologger (Eijkelkamp, Netherland) throughout the 0-80 cm layer with 110 a 30° 2 cm$^2$ cone. In each plot, four sampling zones were randomly selected. In each sampling zone, four penetration measures were performed within an area of 0.25 m$^2$. Disturbed soil samples were also collected to determine gravimetric water content and soil texture in each 20 cm soil layer (0-20, 20-40, 40-60, and 60-80 cm). The penetrologger measured from 0 to 5 MPa. Noteworthy is the fact that the top value was often reached and eventually exceeded in the 60-80 cm layer, although only the 0-60 cm layer was considered in this study. Two PR samplings were performed in the same fashion in the 115 Spring and Autumn surveys as described above, and coincident with the second and third BD measures (Figure 1). PR values



were averaged for each 10 cm of the soil profile and compared with the 2.5 MPa threshold considered a critical value above which root growth may be compromised according to Groenevelt et al. (2001).

### 2.1.3 Saturated hydraulic conductivity and sorptivity

Saturated hydraulic conductivity (Ks) and sorptivity (S) parameters were measured by a double-ring infiltrometer on an area
of 1300 cm$^2$, as described in Morbidelli et al. (2017). Philip's equations (Philip, 1969) were fitted to the field data to calculate Ks and S. Two surveys (spring 2019 and spring 2020) were conducted to measure these parameters after CC termination and before soil preparation.

### 2.2 Statistical analyses

A mixed-effects model was applied to test the main effects of tillage, soil covering, and their interactions on all i-th variables
for each monitoring period. The sand content and bulk density were tested as covariates. All effects named above were treated as fixed effects; the plot effect inside each treatment was treated as random and measurements inside the same plot were considered as nested. All possible first and second order interactions between factors were tested, and the model with the smallest AIC (Akaike's Information Criterion) was selected (Schabenberger and Pierce, 2001).Post hoc pairwise comparisons of least squares means were performed using the Tukey method to adjust for multiple comparisons.
For penetration resistance, the percentage of measures above 2.5 MPa with the whole soil profile considered was tested with Kruskal-Wallis ANOVA, as these data were not-normally distributed. The BD-PR correlation significance was F-tested. All statistical analyses were performed with SAS (SAS Institute Inc. Cary, NC, USA) version 5.1.

## 3 Results

### 3.1 Bulk density

The first BD survey was conducted at the beginning of the experiment (time 0). At that time, BD measurements were uniform among the plots. In particular, BD ranged between 1.14 and 1.60 g cm$^{-3}$ (average value of 1.40 g cm$^{-3}$) in the tilled layer (0-30 cm). In the deepest layer (30-60 cm), the mean value was higher at 1.49 g cm$^{-3}$ within a range of 1.30 g cm$^{-3}$ and 1.69 g cm$^{-3}$. No statistical differences were reported (Fig. 2, Table 1).

**Table 1. Comparison of p values among the linear mixed-effect models analysis of bulk density (BD), penetration resistance (PR),**
**saturated hydraulic conductivity (Ks), and sorptivity (S). Effects were considered significant if p≤0.05.**

|  | | BD | | PR | | Ks | | S | |
|---|---|---|---|---|---|---|---|---|---|
|  | 2018 | Spring | Autumn | Spring | Autumn | 2019 | 2020 | 2019 | 2020 |
| Intercept | 0.0329 | 0.008 | 0.007 | 0.095 | <0.001 | 0.207 | 0.155 | 0.123 | 0.118 |
| Tillage | 0.8849 | <0.001 | 0.003 | <0.001 | 0.034 | <0.001 | <0.001 | <0.001 | <0.001 |




| | | | | | | | | | |
|---|---|---|---|---|---|---|---|---|---|
| CC | 0.0952 | <0.001 | <0.001 | 0.738 | 0.002 | <0.001 | 0.026 | <0.001 | <0.001 |
| Tillage*CC | 0.6640 | <0.001 | <0.001 | 0.006 | 0.014 | <0.001 | <0.001 | <0.001 | <0.001 |
| BULK | # | # | # | 0.280 | 0.369 | -- | -- | -- | -- |
| sand | 0.4293 | <0.001 | 0.573 | <0.001 | 0.041 | 0.2002 | 0.0188 | <0.001 | <0.001 |
| Depth | 0.0000 | <0.001 | <0.001 | <0.001 | <0.001 | # | # | # | # |
| Tillage*Depth | 0.5307 | <0.001 | 0.001 | 0.003 | <0.001 | # | # | # | # |
| CC*Depth | 0.9638 | <0.001 | <0.001 | -- | -- | # | # | # | # |
| Tillage*CC*Depth | 0.9932 | <0.001 | <0.001 | -- | -- | # | # | # | # |
| GWC | # | # | # | 0.404 | 0. 002 | # | # | # | # |

-- effect not included in the model according to the Akaike Information Criterion; # not applicable.

On the contrary, significant differences were reported in the 2020 Spring survey. In the 0-30 cm soil layers, the CT-BS treatment combination displayed the lowest average BD value (1.37 g cm$^{-3}$ or 5.1% lower) among all other treatments. In NT, cover crops TR and WW both seemed to reduce BD values in the 10-40 cm layer (1.54 g cm$^{-3}$ on average) when compared to BS (1.58 g cm$^{-3}$). Generally, a tillage effect was prevalent in the 10-30 cm soil layer (Fig.2). This was demonstrated by a CT average of 1.37 g cm$^{-3}$, as opposed to the 6.5% higher BD values found in the same layer in MT and NT. In the deepest layer, BD values were even higher, ranging from 1.54 g cm$^{-3}$ to 1.91 g cm$^{-3}$. Here, the reduced tillage systems proved to reduce BD moderately, whereas CC produced limited results.

The Autumn BD survey exhibited a greater tillage effect along the soil profile relative to the time-zero survey. Bulk density results in the 0-10 cm layer of NT differed markedly from other treatments. Indeed, they averaged 6.6% above (1.46 g cm$^{-3}$) the others. In these cases, the presence of a cover raised BD values throughout the soil profile by 2.9% (1.41 g cm$^{-3}$). In the subsequent soil layer (10-20 cm), CT showed the lowest average BD values (1.43 g cm$^{-3}$), whereas at depths below 20 cm (20-60 cm), CT treatment resulted in 2.2% higher average BD values (1.57 g cm$^{-3}$) when compared to the reduced tillage systems (MT and NT). Again, the CC effect seemed limited as TR and WW showed 2.8% higher BD (1.48 g cm$^{-3}$) values in the 0-30 cm layer.





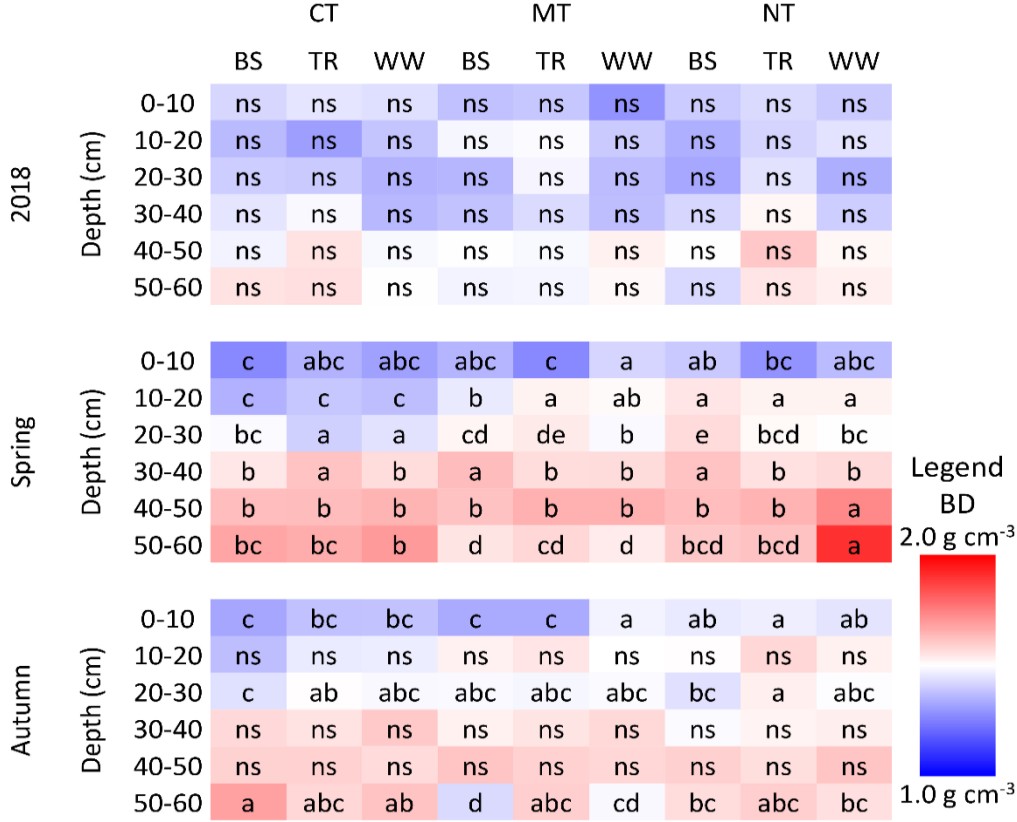

**Figure 2. Bulk density (BD) distribution along the 0-60 cm soil profile.** **For each soil layer, the letters indicate significant effects of tillage x CC according to the Tukey test (p<0.05). CT: conventional tillage; MT: minimum tillage; NT: no-tillage; BS: bare soil; TR: tillage radish; WW: winter wheat.**


### 3.3 Penetration resistance

Results indicated that soil structure, soil texture, and soil water content each affected PR in both 2020 surveys (Table 1). Conditions were, on average, drier during the Autumn survey (0.163 kg kg$^{-1}$) than during the Spring survey (0.222 kg kg$^{-1}$), for which average PR values were 2.52 MPa and 1.58 MPa, respectively. During both surveys, significant tillage × depth and

tillage × CC interactions were detected (Table 1). A comparison among the three tillage systems showed that CT exhibited lower PR values than MT and NT in the 10 to 30 cm depth in both surveys (Fig. 3). Indeed, CT reported an average PR values of 1.04 MPa (Spring) and 1.91 MPa (Autumn), while the reduced tillage treatments increased their PR values +35.6% (1.41 MPa) and +31.4% (2.51 MPa), respectively.




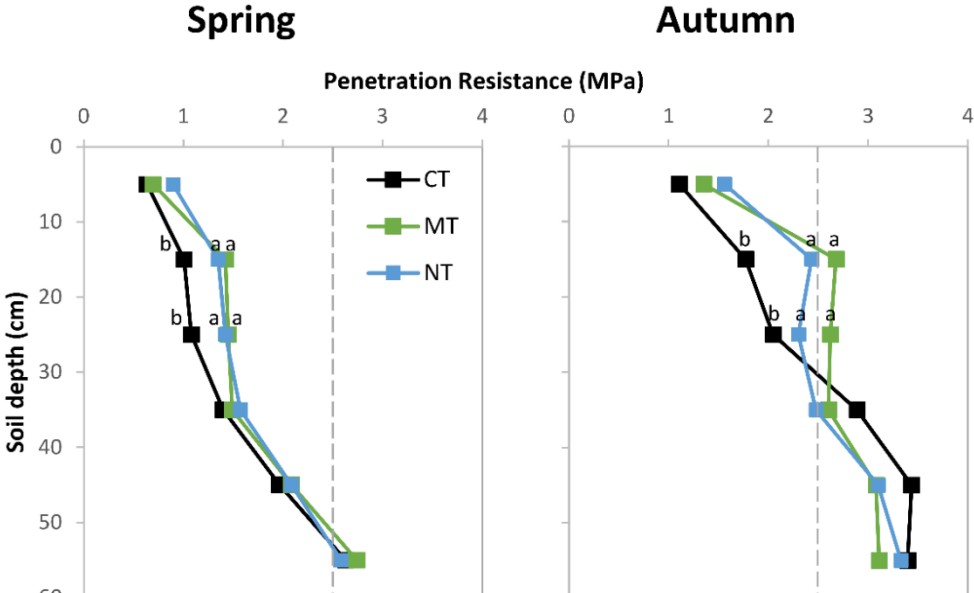

**Figure 3. Penetration resistance (PR) along the 0-60 cm soil profile (values averaged every 10 cm). Different letters represent significant differences according to the post-hoc Tukey test (p<0.05). The vertical dashed line indicates the 2.5 MPa threshold according to Groenevelt *et al.* (2001). CT: conventional tillage; MT: minimum tillage; NT: no-tillage.**

When the entire soil profile was considered, CT (regardless of the winter soil covering), as well as MT-TR and NT-BS were associated with the lowest PR values, in Spring survey (1.50 MPa, on average, Figure 4). The highest PR value occurred in MT-BS (1.74 MPa). Alternatively, in Autumn, the highest PR was measured in MT-TR (2.81 MPa), while MT-BS, CT-WW, CT-BS, and MT-WW (on average 2.42 MPa) were all among the lowest. CT-TR and the NT treatments resulted in intermediate PR values that ranged between 2.51 MPa (NT-WW) and 2.55 (NT-BS).

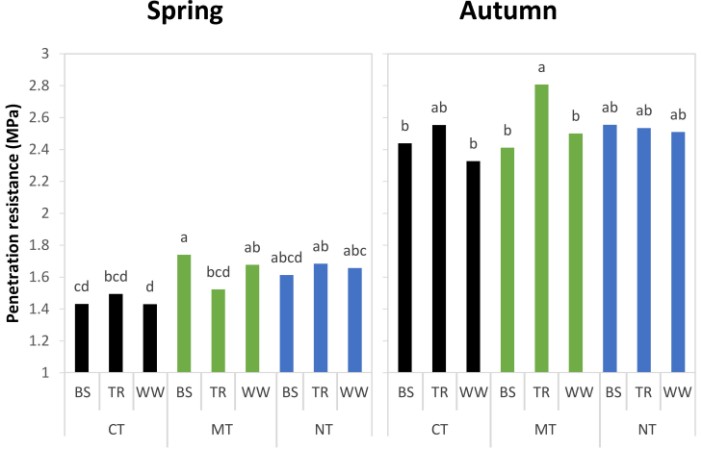


**Figure 4. Penetration resistance along the 0-60 cm soil profile. Different letters represent significant differences according to the post-hoc Tukey test with p<0.05. CT: conventional tillage; MT: minimum tillage; NT: no-tillage; BS: bare soil; TR: tillage radish; WW: winter wheat.**



The PR values were then compared with the 2.5 MPa limit (Fig. 5). During the first survey (Spring) only 13% of measures were above this threshold, mostly beneath the tilled layer. During the Autumn survey, the proportion of measure above the threshold rose to 46%, with a high percentage reported throughout the full soil profile. The Kruskal-Wallis one-way ANOVA indicated there was a significant ($p < 0.05$) effect related to the combination of tillage and CC. Close examination showed that the MT-TR treatment combination resulted with the highest proportion of over-threshold PR values (60%). It

was followed by NT-BS (53%) and all the other treatment combinations ranged between 41% and 45%.



**Figure 5. Percentage of penetration resistance measures above the 2.5 MPa threshold. CT: conventional tillage; MT: minimum tillage; NT: no-tillage; BS: bare soil; TR: tillage radish; WW: winter wheat.**






### 3.4 Soil hydraulic properties

A significant tillage × CC interaction effect was observed on Ks during both the 2019 and 2020 surveys (Fig. 6). The combination of NT-WW produced the highest 2019 Ks value, which represented a two-fold increase compared to all other treatments ($2.50 \times 10^{-5}$ m s$^{-1}$ *vs* $1.04 \times 10^{-4}$ m s$^{-1}$, respectively). During the 2020 survey, all treatments exhibited increased Ks

values that were 1.6 times higher, on average, than those of 2019. In particular, the combination of either BS or WW with NT, had the highest Ks ($2.12 \times 10^{-4}$ m s$^{-1}$), which was more than twice the values of all other treatments ($5.14 \times 10^{-5}$ m s$^{-1}$, on average). It is worth noting that TR displayed no effect in any combination in either year.

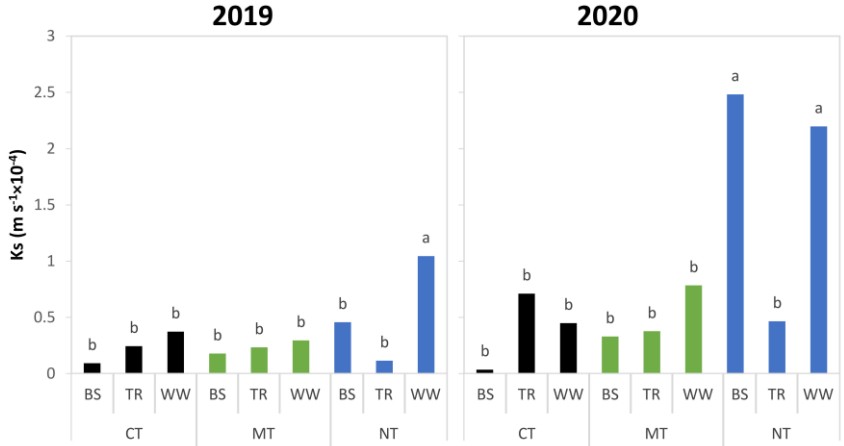

**Figure 6. Saturated hydraulic conductivity (Ks) as measured in the two surveys (2019 and 2020). Different letters represent significant differences according to the post-hoc Tukey test (p<0.05). CT: conventional tillage; MT: minimum tillage; NT: no-tillage; BS: bare soil; TR: tillage radish; WW: winter wheat.**

Sorptivity (S) was affected both by the interaction of tillage × CC and soil texture (Table 1, Fig. 7); sand content negatively

correlated with S. Identical tendencies were observed in both years. Among the various treatment combinations, NT-BS reported the highest results $1.27 \times 10^{-4}$ m s$^{-1}$ (2019) and $3.19 \times 10^{-5}$ m s$^{-1}$ (2020). Very low values of S were observed in CT-BS ($8.5 \times 10^{-7}$ m s$^{-1}$, on average) during the 2020 survey.





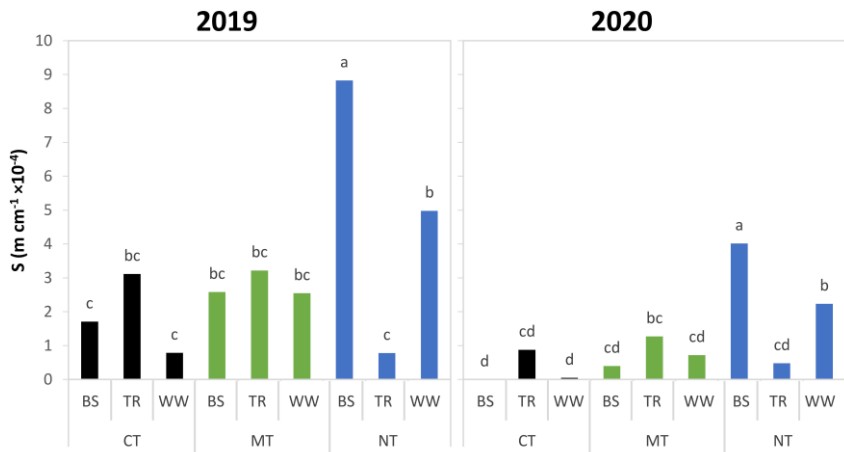

**Figure 7. Sorptivity (S) in the two surveys (2019 and 2020). Different letters represent significant differences according to the post-hoc Tukey test (p<0.05). CT: conventional tillage; MT: minimum tillage; NT: no-tillage; BS: bare soil; TR: tillage radish; WW: winter wheat.**

### 3.5 Correlation between bulk density and penetration resistance

A significant (p<0.01) positive correlation was found between BD (range of 0.5-2.5 MPa) and PR (range of 1.33-1.80 g cm$^{-3}$) with 0.36 $R^2$. At a PR> 2.5 MPa, the correlation with BD was lost; and no other regression could be found between the two parameters. At points above the critical limits of PR (2.5 MPa) and BD (1.55 g cm$^{-3}$), 46% of the observations were detected in CT, 31% in MT, and only 23% in NT, as the red box highlights in Fig. 8. Under these limiting conditions, WW reported the fewest number of measures above this threshold. Following WW was BS; TR had 35% of observations in the range.

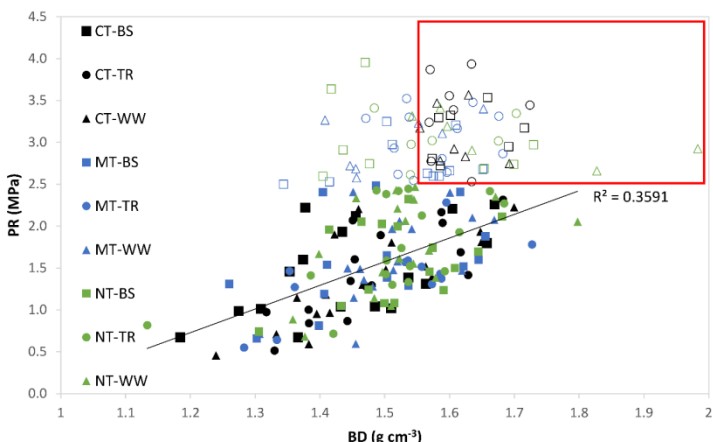

**Figure 8. Correlation between bulk density (BD) and penetration resistance (PR). The line represents the significant (p<0.01) linear correlation for PR<2.5 MPa and BD <1.8 g cm$^{-3}$. Closed and open indicators are used for PRs below or above 2.5 MPa, respectively. The red box highlights observations above both 1.55 g cm$^{-3}$ BD and 2.5 MPa PR.**





## 4. Discussion

The results presented above confirmed that employing a combination of tillage and CC has limited effects in the short term,
as Perego et al. (2019) and Piccoli et al. (2017a) previously reported in similar agroecosystems. Nonetheless, initial, short-term effects on soil physical parameter can be detected in some situations by measuring BD, PR, and soil hydraulic properties. Driven primarily by tillage intensity, lower BD values were found in the tilled layer of both CT and MT. Furthermore, the results highlighted that the magnitudes of BD values at the deeper levels of soil tillage (30 cm ploughing) were similar to those at shallower tillage depths ($\leq$ 15 cm). This finding is consistent with work by Guan et al. (2014).
According to the USDA Natural Resources Conservation Service (1996), a BD value of 1.55 g cm$^{-3}$ in silty loam soils represents a threshold  above which plant growth may be hindered. In this study, this threshold was exceeded, especially at depths below the tilled layer in the first survey (2018), which may be linked to the presence of a plough pan that arose due to repeated soil tillage to the same depth. In a similar agroecosystem, the presence of a plough pan was detected when geophysical and direct assessment methods were combined  (Piccoli et al., 2020). Specifically, the authors found the plough
pan responsible for shallower and greater lateral development of the root apparatus in winter cereals, although it seemed not to affect spring crops (maize, soybean) (Piccoli et al., 2021). During the last survey of the study, both MT and NT exhibited lower BD values beneath the tilled layer. This observation suggests that reduced tillage systems may diminish the strength of a pre-existing hard pan, as is a key goals of CA (Troccoli et al., 2015). Alternatively, given that CC adoption affected BD to only a limited extent, it is quite possible that a longer time period is required to see more change as Blanco-Canqui et al.
(2011) observed in similar pedological conditions. The complexity of the effect of CC on BD as the present study revealed in its 2020 contrasting results from before and after the main cropping season. In fact, seasonal BD changes reported in the literature are generally linked first to meteorological and biological factors (Hu et al., 2012) and secondarily to the time interval after tillage (Ellert and Bettany, 1995; Wendt and Hauser, 2013).

Permeability resistance results confirmed some BD trends. They showed lower average values when associated to wide
differences in tillage intensity (i.e., ploughing *vs* no-tillage). These results agreed with some authors (Trevini et al., 2013) and disagreed with others (Blanco-Canqui and Ruis, 2020; Parihar et al., 2016; Singh et al., 2016). It is worth noting that MT resulted as the tillage with the highest PR values, which contrasted with data obtained in similar pedological conditions, such as Sharratt et al. (2012). As for BD, inconsistent CC results were also found for PR. In general, WW seemed to affect soil strength positively, while TR had either a negligible or negative effect on soil strength. Whereas the well-documented
positive effects of Graminaceous CC on soil physical parameters were expected (Diacono et al., 2019), the inconsistent results for TR were not. In fact, these results were at odds with the reason taproot species were first introduced and adopted as cover crops—for their beneficial effects on soil physical qualities, and soil compaction alleviation, in particular (Toom et al., 2019; Wittwer and van der Heijden, 2020).

The inconsistent results of CC on BD and PR may stem from some methodological issues as well. One such issue is that the
sampling area on which the measures were taken was limited to 39 cm$^2$ for BD and 2 cm$^2$ for PR, whereas the effect from the





apparatus of a taproot cover crop that can only be observed on a larger scale. Another factor may be the various values that authors have suggested as being the PR threshold (de Moraes et al., 2014). It can be hypothesised that under real field conditions, roots can circumvent harder zones if biopores are present. In NT in particular, the high presence of earthworms and the pores left by CC roots—possibly even weed roots—could permit subsequent-crop root penetration into the soil,

despite a high average PR resistance (Hirth et al., 2005).

The analyses of Ks and S highlighted enhanced water infiltration under no-tillage management; moreover, the effects seemed stronger during the second survey. Initially, these results seemed to contrast with BD and PR evidence obtained during the same period. Usually, high BD and PR values are linked to lower soil porosity, so lower Ks values were expected relative to those observed. However, contrasting results on the effects of reduced tillage on Ks also appear in the literature

(Blanco-Canqui and Ruis, 2020; Castellini et al., 2020; Strudley et al., 2008). Indeed, some studies (e.g., Lipiec et al., 2006; Pagliai et al., 2004) have found how the presence of biopores from root decomposition and earthworm activity might alleviate soil compaction by promoting preferential flow through macropores, while other studies (e.g., Kahlon et al., 2013; Vogeler et al., 2009) have suggested that the loss of macroporosity under no-tillage may not sustain water infiltration. The result contrasts such as those observed across the different soil coverings may be influenced by length of the monitoring

period, length of the transition period, and/or issues of scale. A marginal effect that faded during the main cropping season reported amongst the different CC has also been reported by Wagger and Denton (1989). It likely relates to the limited potential of CC to promote well-developed pore networks. Seasonal variability could also have affected soil properties and mask CC effects. Effects from the length of the transition period after conversion from conventional to CA have yet to be fully characterized, although increased soil strength is often observed in the short term. Kay and Vanden Bygaart (2002)

have identified three distinct phases following CA adoption: 1) short-term phase (months): soil compaction and fragmentation is expected from tillage absence and traffic load; 2) medium-term phase (years): greater biological activity (e.g., higher numbers of earthworms) promotes the formation of vertically-oriented bio-macropores, which in turn, alleviates soil strength; 3) extended-term phase (decades): different distributions of soil organic matter stabilize soil structure and fulfil ecosystem servicing needs. In addition, sampling size may also have caused an effect; for example, CC could exert an effect

observable only on a large area (e.g., sub-metric scale), even though most soil analyses (e.g., bulk density) are performed at smaller scales (e.g., centimetre-scale) (Piccoli et al., 2019).

In this study, the presence of a BD-PR correlation capable of depiction only in the 0.5-2.5 MPa and 1.33-1.80 g cm$^{-3}$ ranges may suggest that in lower density soil profiles (i.e., BD<1.8 g cm$^{-3}$ and PR<2.5 MPa), soil structure dynamics might be governed by a centimetre scale due to a homogeneous pore network. On the contrary, higher density (e.g., BD>1.8 g cm$^{-3}$

and PR>2.5 MPa) soils might be characterized by low anisotropic porosity, in which the presence/absence of few macropores (e.g., cracks, biopores) may rule structure dynamics and soil functions in the form of water infiltration and/or gas exchanges (Piccoli et al., 2017a, 2019). The inconsistent results seen in no-tillage systems probably were caused by a scale issue as well. Indeed, NT evidenced soil compaction and satisfactory water infiltration simultaneously, likely due to the



presence of vertically-oriented biomacropores and greater pore connectivity (Piccoli et al., 2017b). Consequently, both CC
and NT systems are likely to produce more heterogeneous soil structure as compared with tilled soils.

**Conclusions**

This study proved that during the transition period from conventional to conservation agriculture some compaction issues
can be linked to no-tillage when monitoring is performed with traditional small-scale physical methods (e.g., bulk density,
penetration resistance) due to a high soil structure heterogeneity. Therefore, the use of larger-scale measurement, such as the
double ring infiltrometer, might be preferable in no-tillage managements to overcome the inherent problems of higher spatial
variability at the micro scale and to consider soil function as a whole. The fibrous root apparatus of *Poaceae* species seems a
promising cover crop to enhance soil physical qualities in the no-tillage systems of Northeast Italy, even in the short term.
Moreover, Graminaceous, such as winter wheat, are commonly cash crops in this study area and their agronomic
management (e.g., sowings) is easily implemented by farmers. However, the longer period required for taproot cover crop
(e.g., tillage radish), and no-till systems alike, to exploit its ecosystem services fully requires their evaluation at a larger
scale. One of the future challenges that the agronomic community will face is the termination of cover crops, especially in
light of pesticide reduction, and/or the selection of winter-killed species to meet the sustainable development goals of the
2030 Agenda.

*Data availability.* The data that support the findings of this study are available from the corresponding author upon
reasonable request.

*Author contributions.* Conceptualization, A.B.; methodology, I.P., A.B.; formal analysis, F.S.; investigation, F.S. and R.P.;
Resources, R.P.; data curation, F.S. and R.P.; writing—original draft preparation, F.S. and I.P.; writing—review and editing,
F.S., I.P., R.P. and A.B.; visualization, F.S.; supervision, A.B. All authors have read and agreed to the published version of
the manuscript.

*Competing interests.* The authors declare that they have no conflict of interest.

*Funding.* The research leading to these results has received funding from the European Union HORIZON2020 Programme
for Research, Technological Development, and Demonstration under Grant Agreement No. 677407 (SOILCARE Project).

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
