# Peer review of "Transition to conservation agriculture: how tillage intensity and covering affect soil physical parameters"

_SOIL, 2021_

## Author Comment (AC1)

**RC1 Anonymous Referee #1, 21 Oct 2021**

| *RC1 comments* | *Authors' answers* |
|---|---|
| In this study the short-term effects of conservation agriculture practices evaluated for their effect in three soil physical properties. Different soil cultivation and soil cover treatments evaluated for their effects on bulk density, penetration resistance and hydraulic conductivity + sorptivity. The authors used mixed effects models to analyse the effects and interactions. The outcome of the research is highly relevant to improve the existing knowledge on CA and promote proper adoption of CA practices in the region of the study site.

The authors have used a proper experimental design and experimental procedures combined with state of the art and advanced statistical analysis. The manuscript though, needs language refinement and additions especially in the introduction section and results presentation to achieve an excellent overall quality. | Thank you. This comment helped us to give important context to the reader in the Introduction. We both modified and made the terminology consistent to avoid any misunderstandings. We took care to consider each observation and integrate it considered each observation and used it to improve the manuscript.

The whole text was revised by a professional English reviewer, who also reviewed the edited version to ensure linguistic correctness. |
| **Specifically:** | |
| In the introduction, in the first paragraph the benefits and drawbacks of CA should be added, coupled with results from existing literature. In the third paragraph where the situation in Italy is described the half paragraph is about general drawbacks and benefits of CA and is more suitable to be moved in the first one. The fourth paragraph describes the suitable species and situation for Italy and should be merged with the third one. Also, the tillage systems used in Italy should be mentioned. | We have clarified the Introduction as directed. Specifically, we clarified both the positive and negative results we observed and provided additional description about Northern Italy tillage practices. |
| In the 5[th] paragraph you mention that these measurements cover different spatial resolutions, but these measurements quantify different soil physical properties. It is not a matter of scale but a matter of different properties, and this should be clarified and corrected in the text. I think you should reconsider your scientific question. | We agree that the measured properties quantify different soil properties. However, these properties are usually correlated at the plant/field scale. A soil with high hydraulic conductivity is usually less dense and strong, as compared to soils with lower Ks. If these soil properties seem uncorrelated, then there might be a scale issue. Recent literature stresses the importance of considering the proper representative elementary volume in soil |

| | analyses. Point measurement, such as with penetration resistance, clearly showed the presence of hard horizons in the soil, but under field conditions, roots can bypass these horizons if there is cracking, biopores, channels left by degraded roots, and so on. Given this, we think that measurements made on a larger scale, such as water infiltration, can better mimic root behaviour, as it is affected by the presence of channels that roots can use for their growth. So, from a root growth perspective, considering a wider area seemed more appropriate. Still, we understand that this aspect needs clarification, so we modified our text at the end of the Introduction and Discussion sections. |
|---|---|
| In the methods section clarify the experimental design. In L 88 I suppose you mean in strips not in plots Be careful with the terms. In a split plot the whole plot is split into subplots (or strips) and the first factor is allocated there- I suppose is tillage for you- and then the second factor is randomly allocated within these in the experimental units. So, I think you have 18 experimental units. Please use the proper terminology throughout the manuscript. It would be nice to include the experimental design layout as a figure. | We understand that this aspect requires clarification. In fact, we modified the text to clarify that the main treatment consists of three different tillage intensities. These different managements were applied in a randomized main plot within each block. Subsequently, each main plot was divided into three subplots. Each subplot received a different soil covering management. |
| For the surveys you should add months also in Figure 1 to give a perspective of time within the year. And also specify the replicates per experimental unit (within the plot replicates) for all the measurements. Eg how many BD undisturbed samples you collected from each experimental unit. | We redesigned the figure accordingly. Then, we clarified both the timing and replicates of each sampling in the text. |
| Finally indicate the p value in the method. | We added the exact p value to the tables for each ANOVA comparison. |
| In the results you refer to texture measurements, effects and correlation without presenting the variation of texture within the plots. | The differences in soil texture among the plots were limited and not significant. We emphasized will this point and added more information on texture and its variability. |
| **Detailed comments:** | |
| L6: CA relies in three main piles add also crop rotation | We modified accordingly. |
| L7 and other places in the text: Correct soil physics to soil physical benefits or soil physical properties. Soil physics is the science and it include a wide range of properties and | Thank you for the comment. We made the modifications. |

| concepts | |
|---|---|
| L7: is reduced soil strength a benefit? | Average soil strength results were high in compacted soil. Its reduction may alleviate this threat. |
| L10: Define BD, PR in parenthesis and other abbreviations the first time appear in both in abstract and introduction before you use the short versions | We modified accordingly. |
| L:10 and other places in the text: Change measures to measurements. Be careful when used measurements: the quantification of attributes of an object or event e.g. measurements of BD, weight etc. Measures: actions taken to achieve a particular purpose e.g. no tillage cover crops etc | Thank you for the comment. We modified as suggested. |
| L10: Define what soil hydraulic measures | We added this information. |
| L10: To evaluate the soil quality not the results | We modified accordingly. |
| L13: use more or other word instead of better | We used "better" because the soil is compacted and a reduction in BD could be considered a better condition. We changed the terminology accordingly. |
| L13: define or the percentage change in parenthesis or write from how much reduced to the second value | We modified accordingly. |
| L15: see comment for line L7 | We modified accordingly. |
| L15-16 "as evidenced by root growth-limiting threshold declines (-11% in BD values >1.55 g cm-3 and -7% in PR values >2.5 MPa)." Rephrase | We rephrased this sentence for clarity. |
| L16: define what measure not only soil hydraulic measurements | We add this information. |
| L20: specify how the strategy enhances soil physical properties | We rephrased this sentence. |
| L21: change to "This study demonstrates that to quantify CA effects requires monitoring several soil physical parameters." or similar | We modified accordingly. |
| L25-28: references needed | We add the reference HOBBS, Peter R.; SAYRE, Ken; GUPTA, Raj. "The role of conservation agriculture in sustainable agriculture. Philosophical Transactions of the Royal Society B": Biological Sciences, 2008, 363.1491: 543-555. |
| L28: specify what type of contrasting results have been reported | We added details to the results of the contrasts reported for this source that analysed different soil physical parameters. |
| L30: reference is needed | This sentence shares the reference of the previous line. We rephrased these two sentences to add more detail and to specify the |

| | references clearly. |
|---|---|
| L39-40: What situations?  specify | We were referring to specific conditions, such as the presence of a hardpan, high weed pressure, or the needs of slurry managements. We clarified all. |
| L66- L 95: the BD and PR have already been used before. Specify only the first time mentioned in the text. | We modified accordingly. |
| L81 and other places in the text: Change rainfalls to rainfall. | We modified accordingly. |
| L107: specify the volume and height of the core and give details for the sampling depths (0-20, 20-40 etc). and how many cores per depth and per experimental plot. | We specified the information more completely. |
| L110: Do you mean experimental units? | We enhanced the description of the experimental design by underlining that within each plot we had four sampling zones. I.e.: 4 sampling zones x 3 soil cover managements x 3 tillage managements x 2 blocks = 72 total sampling zones. Within each sampling zone, we collected disturbed soil samples and performed four penetration measurements. |
| L111: change to measurements | We modified accordingly. |
| L112-114: I believe this belong to the results | We moved this section to the Results. |
| L116: threshold which is considered | We modified accordingly. |
| L119: You measure infiltration rates and from that you calculated the Ks and S with the Philips equation please change. | We clarified this question. |
| L121: Indicate the number of within the experimental plot replicates of the measurement | We clarified this question. |
| L126: the plot effect – remove inside each treatment. | We clarified this paragraph, by adding complete and clear information according to reviewers' comments. |
| L130:do you mean within the whole profile? | |
| L136: The DB range may not be significant statistically but is important physically. You should elaborate on the impacts of these values. | We evaluated summarizing all the data into a table. It could be useful to have more information, even if there are no significant differences. On the contrary, and based on other comments, it seems important to keep the Results section simple to avoid misunderstandings. |
| Table 1 change the captions/ It is not easy for the user to figure out the sampling when half of these are seasons and the other half years. Use uniform format. E.g. spring 2018 and also specify in the text why you had no applicable # (e.g. measurements only on the topsoil) Also in the first column use same format for the | We clarified this point by adding the years to the Table as suggested. |

| words. Some are only capital letters other start with capital etc. Specify what is GWC | |
|---|---|
| 139 and many other places in the text: Some times you use Figure in the main text to refer to the figure and some other Fig. Please use the same format. | We standardized this. |
| L 189 and other places in the text remove the word combination next to treatment as by default the treatments is a combination of factors. So, either use for example the MT-TR treatments or the MT-TR combination | We modified accordingly. |
| L189: resulted in | We modified accordingly. |
| L199 you use respectively but you dop not refer to which treatments | We clarified that the first data is referred to as NT-WW and the other refers to the average values of all other treatments (which were not significant differences). |
| L 219 change the word lost with a more suitablke | We rephrased. |
| L222 above which of the two thresholds? Or you mean these instead of this? | We corrected to "these" thresholds. |
| L223 which range you mean please specify | We changed the text to be "above the two thresholds", which was our intention. |
| L226 what do you mean by closed or open indicators? I think you mean solid and symbols | We meant open symbols and closed symbols. We modified accordingly. |
| L: 229 Which results specifically and effects on what? | We rephrased for clarification. |
| L231: effects on soil physical properties or soil physical condition | We rephrased to clarify. We found significant differences in the observed soil physical parameters starting with the first years of conversion from conventional tillage to conservation agriculture. |
| L243 wrong syntax | We revised this sentence. |
| L251-252 and many places in the discussion specify what these authors found instead of only mention the names. Eg The results aggree with XX who found .... and disagree with xx who fount .... | We added this information. |
| L300 as mentioned before these measurements are used to evaluate different soil properties. You should not compare their scale. In order to reduce the effect of soil heterogeneity you replicate the same measurement within each experimental plot more times. You cannot say that by using the infiltrometer which measure infiltration capacity can overcome the variability problems you face when measure BD just because it covers a bigger area. These are two different | We clarified this. As the reviewer correctly highlighted, the three methods considered different soil physical properties. Nevertheless, all of them provide information on soil function and soil root habitability, especially within the context of poorly-structured soil and the threat of soil compaction. Consequently, we argued that the different results could be related to the scale difference. We did not intend to suggest that the infiltrometer could replace BD or PR. As the double-ring infiltrometer investigated a |

| unique measurements.  I think you should reconsider/remove that part | wider soil portion, it seemed to take spatial variability into account better. In fact, while PR and BD seemed to be negatively correlated with reduced tillage system adoption, the infiltrometer produced opposite results. That is, NT had the highest Ks values, which resulted in a positive impact from the reduced tillage system on soil hydraulic properties.

We clarified the text to avoid any misunderstanding. |
|---|---|

https://doi.org/10.5194/soil-2021-113-RC1

---

## Author Comment (AC2)

| **CC1: Marta Diaz, 04 Jan 2022** |  |
|---|---|
| *RC1 comments* | *Authors' answers* |
| The present study evaluates the effects of conservation agricultural practices, focusing on the effects in three specific soil physical properties. The results obtained in this study are relevant and could improve the future implementation of conservation agriculture.

Overall, the authors have carried out a very good job in the design and writing of this manuscript. However, some modifications should be made to achieve a manuscript of high scientific quality. The following are some suggestions for modifications to the manuscript. | We would like to express our sincere thanks for this comment, which will help us to improve our manuscript. |
| Q1. Restructure the Introduction part so that it has a cohesive and consistent thread. | We will revise this section, in light of all the comment we received. |
| Q2. The description of the methods in the Introduction is a bit confusing because the authors describe them as scales and not as measured properties. | We will revise this part, especially considering Anonymous Referee #1 comments. We will clarify that the different methods provide different information, but in the studied soil, all of them are measure of soil root habitability and soil function, and different results obtained from the three different parameters could be related to the different scale and resolution of the methods. |
| Q3. Material and methods section. Why were not all the soil physical properties analyzed at the same time? It is confusing. | We will try to clarify the timing of the sampling. We considered valuable to have more data after three-year conversion, especially before and after the main crop. Nevertheless, the timing of these measurements is subjected to many factors, such as soil moisture, field accessibility, weather that could change the planning. |
| Q4. The Results section is a bit difficult to understand. I would recommend detailing only the most important results, followed by the corresponding p-value. | According to this and other comment, we will revise this part, clarifying the results while providing complete information |
| Q5. The results obtained in this assay do not appear to be consistent with the results obtained in other assays. However, the authors do not specify how they differ from the results available in the literature. To enrich the Discussion part, it would be desirable for the authors to discuss more the differences with already published results and possible | We will provide more details on other authors findings, as suggested. |

| | |
|---|---|
| hypotheses that could explain these differences, rather than just highlighting that differences exist. | |
| Q6. Define abbreviations (BD, PR...) the first time they appear before using them. | We will modify accordingly. |
| Q7. Figure 1. Add months to the timeline to make it easier to understand the essay. | We will modify accordingly. |

https://doi.org/10.5194/soil-2021-113-CC1

---

## Author Comment (AC3)

| **RC2: Anonymous Referee #2, 11 Jan 2022** | |
|---|---|
| *RC1 comments* | *Authors' answers* |
| In this study, the authors have evaluated how the conversion of conventional agriculture to conservation agriculture could affect soil physical properties. For this purpose, the authors have monitored different soil physical properties during 3 years in plots with different tillage treatments and different cover crops. These soil physical properties were the bulk density (BD), penetration resistance (PR), hydraulic conductivity (Ks), and sorptivity (S). The work results showed that the absence of tillage enhances soil physical properties. At the same time, the use of some cover crops also improves the soil physics. In general, the research makes sense since it looks to increase the knowledge about the effects caused in the soil during the transition to conservative agriculture. However, the manuscript needs a few improvements before its publication. Some parts of the text are a little difficult to read. The experimental method could be clarified to improve its understanding. Moreover, in the results section, there is too much information in parentheses. I would recommend only writing the necessary numeric values to well describe the work results. Some parts of the text should be rewritten to do it more readable and intelligible. Finally, the part of references shows some little mistakes. I specify them below. Please, correct them. | We thank the reviewer for the precious comments. We will improve the manuscript accordingly. Particularly, we will better describe the methods as observed by all the reviewers, we will also clarify the results, highlighting the significant difference while summarizing the other information. The references will be carefully revised to avoid inhomogeneity. Finally, the whole manuscript was revised by a professional English reviewer and will be revised after the editing to guaranty the language correctness and clarity. |
| L10. I would recommend to write the short version of bulk density and penetration resistance in parenthesis the first time that appear in the text. | We will modify accordingly. |
| L10. I consider that 'soil hydraulic measures' is unspecific. I would recommend to be more specific when writing an abstract. Please, change this to 'saturated hydraulic conductivity (Ks) and sorptivity (S)'. | We will clarify accordingly. |
| L25-28. Please, add some references that support it. | We will add this reference: HOBBS, Peter R.; SAYRE, Ken; GUPTA, Raj. The role of conservation agriculture in sustainable agriculture. Philosophical Transactions of the Royal Society B: Biological Sciences, 2008, 363.1491: 543-555. |

| L74-77. There was no hypothesis in the sentences where you defined the aims of the research. What were the expectations for this research? What results did you expect to obtain? On what previous evidences were based your expectations? | We will clarify the hypothesis: the introduction of reduced tillage system was expected to negatively impact on the studied soil physical properties, but the combination of reduced tillage system with tillage radish was expected to alleviate these drawbacks. This hypothesis results partially rejected as only some of the studied parameter resulted negatively affected by CA introduction, while TR seemed to have limited impact. |
|---|---|
| L95. BD and PR have been already used before (L66-77). Write in parentheses only the first time you mentioned. | We will modify accordingly. |
| Figure 1. Why were not bulk density and penetration resistance analysed in 2019? | Both BD and PR are invasive tests and excessive repetitions could impact on soil structure. BD particularly requires heavy machinery which could cause soil compaction, while penetration resistance was performed with many replicates, which results in soil disturbance. Thus, we retained more important to have two measures in the last experimental year, to monitor the evolution of these parameters along a single growing season, when the first effects of conversion to CA a should start to be evident. In fact, literature reports long conversion time and often in the first years negative and positive CA effect are not easily assessable. |
| L114-115. Why was not the penetration resistance analysed in 2018 (time 0)? Please, explain it. | As mentioned in the first comment, all the sampling we performed are destructive, and required specific pedoclimatic conditions together with field accessibility, and absence of the main culture. Particularly PR required enough soil moisture, and the studied soil results often too dry for this analysis. |
| L128-129. Were the normality and homoscedasticity of data checked? Please, specify it. | We tested these properties. We will add this information. |
| L139-140. Define GWC in the Table 1 caption. | We will revise accordingly. |
| L165-168. It would be interesting to know if there were differences in the penetration resistance among the different cover crop for each tillage treatments every 10 centimetres along the soil profile. Were these differences analysed? If affirmative, were significant these differences? | No significant differences were founded between these treatments. We will explain this finding. |
| L325. The reference is not correct. The name of authors and the year of publication are missing. | We will revise this reference. |

| | |
|---|---|
| Please, correct it. | |
| L356. The DOI appears twice. Please, correct it. | We will remove the repetition. |
| L363. The DOI is missing. Please, correct it. | We will add this information. |
| L406. See comment for line L363. | We will add this information. |
| L429. See comment for line L363. | We will add this information. |

https://doi.org/10.5194/soil-2021-113-RC2

---

## Author Response (AR1)

| **RC1 Anonymous Referee #1, 21 Oct 2021** | |
|---|---|
| *RC1 comments* | *Authors' answers (line number refers to the PDF with track changes)* |
| In this study the short-term effects of conservation agriculture practices evaluated for their effect in three soil physical properties. Different soil cultivation and soil cover treatments evaluated for their effects on bulk density, penetration resistance and hydraulic conductivity + sorptivity. The authors used mixed effects models to analyse the effects and interactions. The outcome of the research is highly relevant to improve the existing knowledge on CA and promote proper adoption of CA practices in the region of the study site.

The authors have used a proper experimental design and experimental procedures combined with state of the art and advanced statistical analysis. The manuscript though, needs language refinement and additions especially in the introduction section and results presentation to achieve an excellent overall quality. | Thank you. This comment helped us to give important context to the reader in the Introduction. We both modified and made the terminology consistent to avoid any misunderstandings. We took care to consider each observation and integrate it considered each observation and used it to improve the manuscript.

The whole text was revised by a professional English reviewer, who also reviewed the edited version to ensure linguistic correctness. |
| **Specifically:** | |
| In the introduction, in the first paragraph the benefits and drawbacks of CA should be added, coupled with results from existing literature. In the third paragraph where the situation in Italy is described the half paragraph is about general drawbacks and benefits of CA and is more suitable to be moved in the first one. The fourth paragraph describes the suitable species and situation for Italy and should be merged with the third one. Also, the tillage systems used in Italy should be mentioned. | We have clarified the Introduction as directed. Specifically, we clarified both the positive and negative results we observed and provided additional description about Northern Italy tillage practices (LL68-73). Moreover the entire introduction section was revised, please see LL30-66, LL70-92 and LL103-121. |
| In the 5[th] paragraph you mention that these measurements cover different spatial resolutions, but these measurements quantify different soil physical properties. It is not a matter of scale but a matter of different properties, and this should be clarified and corrected in the text. I think you should reconsider your scientific question. | We agree that the measured properties quantify different soil properties. However, these properties are usually correlated at the plant/field scale. A soil with high hydraulic conductivity is usually less dense and strong, as compared to soils with lower Ks. If these soil properties seem uncorrelated, then there might be a scale issue. Recent literature stresses the importance of considering the proper |

| | representative elementary volume in soil analyses. Point measurement, such as with penetration resistance, clearly showed the presence of hard horizons in the soil, but under field conditions, roots can bypass these horizons if there is cracking, biopores, channels left by degraded roots, and so on. Given this, we think that measurements made on a larger scale, such as water infiltration, can better mimic root behaviour, as it is affected by the presence of channels that roots can use for their growth. So, from a root growth perspective, considering a wider area seemed more appropriate. Still, we understand that this aspect needs clarification, so we modified our text at the end of the Introduction and Discussion sections. |
|---|---|
| | We wish to highlight that the entire section has been reworded, please see LL36-121. |
| In the methods section clarify the experimental design. In L 88 I suppose you mean in strips not in plots Be careful with the terms. In a split plot the whole plot is split into subplots (or strips) and the first factor is allocated there- I suppose is tillage for you- and then the second factor is randomly allocated within these in the experimental units. So, I think you have 18 experimental units. Please use the proper terminology throughout the manuscript. It would be nice to include the experimental design layout as a figure. | We understand that this aspect requires clarification. In fact, we modified the text to clarify that the main treatment consists of three different tillage intensities. These different managements were applied in a randomized main plot within each block. Subsequently, each main plot was divided into three subplots. Each subplot received a different soil covering management.

 We clarified the experimental design in LL133-143. |
| For the surveys you should add months also in Figure 1 to give a perspective of time within the year. And also specify the replicates per experimental unit (within the plot replicates) for all the measurements. Eg how many BD undisturbed samples you collected from each experimental unit. | We redesigned the figure accordingly. Then, we clarified both the timing and replicates of each sampling in the text see Figure 1 and e.g., LL153, 158 and 177. |
| Finally indicate the p value in the method. | We added the p value in L189. |
| In the results you refer to texture measurements, effects and correlation without presenting the variation of texture within the plots. | The differences in soil texture among the plots were limited and not significant. We emphasized added this information in LL 134-135. |
| **Detailed comments:** | |
| L6: CA relies in three main piles add also crop | We modified the abstract accordingly, in L 6. |

| rotation | |
|---|---|
| L7 and other places in the text: Correct soil physics to soil physical benefits or soil physical properties. Soil physics is the science and it include a wide range of properties and concepts | Thank you for the comment. We made the modifications, see e.g., 7, 17 and 345. |
| L7: is reduced soil strength a benefit? | Yes, reduced soil strength is a benefit in terms of compaction mitigation. |
| L10: Define BD, PR in parenthesis and other abbreviations the first time appear in both in abstract and introduction before you use the short versions | We modified accordingly, see e.g., L10, 11 and 13. |
| L:10 and other places in the text: Change measures to measurements. Be careful when used measurements: the quantification of attributes of an object or event e.g. measurements of BD, weight etc. Measures: actions taken to achieve a particular purpose e.g. no tillage cover crops etc | Thank you for the comment. We modified as suggested, e.g., L18, 166 and 171. |
| L10: Define what soil hydraulic measures | We added this information in L11. |
| L10: To evaluate the soil quality not the results | We modified accordingly in LL11-12. |
| L13: use more or other word instead of better | We used "better" because the soil is compacted and a reduction in BD could be considered a better condition. We changed the terminology accordingly in L15. |
| L13: define or the percentage change in parenthesis or write from how much reduced to the second value | We modified accordingly in LL14-17. |
| L15: see comment for line L7 | Yes, we confirm that having soils with BD and PR below the growth-limiting threshold is positive for correct crop growth. |
| L15-16 "as evidenced by root growth-limiting threshold declines (-11% in BD values >1.55 g cm-3 and -7% in PR values >2.5 MPa)." Rephrase | We rephrased this sentence for clarity in L16. |
| L16: define what measure not only soil hydraulic measurements | The specific hydraulic measurements are already specified in L19 and 20. |
| L20: specify how the strategy enhances soil physical properties | We rephrased this sentence please see LL22-23. |
| L21: change to "This study demonstrates that to quantify CA effects requires monitoring several soil physical parameters." or similar | We modified accordingly in LL24-25. |
| L25-28: references needed | We add the reference HOBBS, Peter R.; SAYRE, Ken; GUPTA, Raj. "The role of conservation agriculture in sustainable agriculture. Philosophical Transactions of the Royal Society B": Biological Sciences, 2008, 363.1491: 543-555 in LL31-32. |

| | |
|---|---|
| L28: specify what type of contrasting results have been reported | We added details to the results of the contrasts reported for this source that analysed different soil physical parameters in LL40-75. |
| L30: reference is needed | The entire section has been completely modified, please see LL43-48 |
| L39-40: What situations? specify | We were referring to specific conditions, such as the presence of a hardpan, high weed pressure, or the needs of slurry managements. We clarified all. Nevertheless, since the section has been completely revised, this part has been deleted for a more text fluency. |
| L66- L 95: the BD and PR have already been used before. Specify only the first time mentioned in the text. | We modified accordingly. |
| L81 and other places in the text: Change rainfalls to rainfall. | We modified accordingly in LL126-128. |
| L107: specify the volume and height of the core and give details for the sampling depths (0-20, 20-40 etc). and how many cores per depth and per experimental plot. | We specified the information more completely in L155-159. |
| L110: Do you mean experimental units? | We enhanced the description of the experimental design by underlining that within each plot we had four sampling zones. I.e.: 4 sampling zones x 3 soil cover managements x 3 tillage managements x 2 blocks = 72 total sampling zones. Within each sampling zone, we collected disturbed soil samples and performed four penetration measurements. Please see L136-139. |
| L111: change to measurements | We modified accordingly. |
| L112-114: I believe this belong to the results | We moved this section to the Results, please see LL226-227. |
| L116: threshold which is considered | We modified accordingly in L172. |
| L119: You measure infiltration rates and from that you calculated the Ks and S with the Philips equation please change. | We clarified this question in L175. |
| L121: Indicate the number of within the experimental plot replicates of the measurement | We clarified this question in LL178-179. |
| L126: the plot effect – remove inside each treatment. | We revised the entire section, for clarity, see LL181-191. |
| L130:do you mean within the whole profile? | |
| L136: The DB range may not be significant statistically but is important physically. You should elaborate on the impacts of these values. | We evaluated summarizing all the data into a table. It could be useful to have more information, even if there are no significant differences. On the contrary, and based on other comments, it seems important to keep the Results section simple to avoid |

| | misunderstandings. |
|---|---|
| Table 1 change the captions/ It is not easy for the user to figure out the sampling when half of these are seasons and the other half years. Use uniform format. E.g. spring 2018 and also specify in the text why you had no applicable # (e.g. measurements only on the topsoil) Also in the first column use same format for the words. Some are only capital letters other start with capital etc. Specify what is GWC | We clarified this point by adding the years to the Table as suggested, please see Table 1. |
| 139 and many other places in the text: Some times you use Figure in the main text to refer to the figure and some other Fig. Please use the same format. | We standardized this. |
| L 189 and other places in the text remove the word combination next to treatment as by default the treatments is a combination of factors. So, either use for example the MT-TR treatments or the MT-TR combination | We modified accordingly, see e.g., 205, 256 and 257. |
| L189: resulted in | We modified accordingly in L256. |
| L199 you use respectively but you dop not refer to which treatments | We clarified that the first data is referred to as NT-WW and the other refers to the average values of all other treatments (which were not significant differences), see L267. |
| L 219 change the word lost with a more suitablke | We rephrased in L290. |
| L222 above which of the two thresholds? Or you mean these instead of this? | We rephrased the sentence in LL292-294. |
| L223 which range you mean please specify | We rephrased the sentence in LL292-294. |
| L226 what do you mean by closed or open indicators? I think you mean solid and symbols | We meant open symbols and closed symbols. We modified accordingly in the new caption of Fig.2. |
| L: 229 Which results specifically and effects on what? | We rephrased the entire section for a better clarity, please see LL300-309. |
| L231: effects on soil physical properties or soil physical condition | We rephrased to clarify in L305. |
| L243 wrong syntax | We revised this sentence in L318. |
| L251-252 and many places in the discussion specify what these authors found instead of only mention the names. Eg The results aggree with XX who found .... and disagree with xx who fount .... | Following the reviewer's comment we revised the entire section, please see e.g., L301-304, LL325-327 and 339-344. |
| L300 as mentioned before these measurements are used to evaluate different soil properties. You should not compare their scale. In order to reduce the effect of soil heterogeneity you replicate the same measurement within each experimental plot | We clarified this. As the reviewer correctly highlighted, the three methods considered different soil physical properties. Nevertheless, all of them provide information on soil function and soil root habitability, especially within the context of poorly-structured soil and the threat |

more times. You cannot say that by using the infiltrometer which measure infiltration capacity can overcome the variability problems you face when measure BD just because it covers a bigger area. These are two different unique measurements. I think you should reconsider/remove that part

of soil compaction. Consequently, we argued that the different results could be related to the scale difference. We did not intend to suggest that the infiltrometer could replace BD or PR. As the double-ring infiltrometer investigated a wider soil portion, it seemed to take spatial variability into account better. In fact, while PR and BD seemed to be negatively correlated with reduced tillage system adoption, the infiltrometer produced opposite results. That is, NT had the highest Ks values, which resulted in a positive impact from the reduced tillage system on soil hydraulic properties.

We clarified the entire section for better understanding. See LL381-387

https://doi.org/10.5194/soil-2021-113-RC1

| **CC1: Marta Diaz, 04 Jan 2022** | |
| --- | --- |
| *RC1 comments* | *Authors' answers* |
| The present study evaluates the effects of conservation agricultural practices, focusing on the effects in three specific soil physical properties. The results obtained in this study are relevant and could improve the future implementation of conservation agriculture.

Overall, the authors have carried out a very good job in the design and writing of this manuscript. However, some modifications should be made to achieve a manuscript of high scientific quality. The following are some suggestions for modifications to the manuscript. | We would like to express our sincere thanks for this comment, which helped us to improve our manuscript. |
| Q1. Restructure the Introduction part so that it has a cohesive and consistent thread. | We revised the entire section, as requested. See LL27-121 |
| Q2. The description of the methods in the Introduction is a bit confusing because the authors describe them as scales and not as measured properties. | We revised this part, especially considering Anonymous Referee #1 comments. See LL103-121. |
| Q3. Material and methods section. Why were not all the soil physical properties analyzed at the same time? It is confusing. | We try to clarify the timing of the sampling. We considered valuable to have more data after three-year conversion, especially before and after the main crop. Nevertheless, the timing of these measurements is subjected to many factors, such as soil moisture, field accessibility, weather that could change the planning. However we clarify better the time of sampling in Table 1 and Figure 1 and in many places in the text e.g., 175-179. |
| Q4. The Results section is a bit difficult to understand. I would recommend detailing only the most important results, followed by the corresponding p-value. | According to this and other comment, we revised this part, clarifying the results while providing complete information, see e.g., 195-196, 210-213, 216-218, 226-227, 234, 255-256, 267 and 270-271. |
| Q5. The results obtained in this assay do not appear to be consistent with the results obtained in other assays. However, the authors do not specify how they differ from the results available in the literature. To enrich the Discussion part, it would be desirable for the authors to discuss more the differences with already published results and possible hypotheses that could explain these differences, rather than just highlighting that | According to reviewer's comment we modified the entire section, see e.g., LL301-304, 325-328 and 338-344. |

| | |
|---|---|
| differences exist. | |
| Q6. Define abbreviations (BD, PR...) the first time they appear before using them. | We modified accordingly. |
| Q7. Figure 1. Add months to the timeline to make it easier to understand the essay. | We modified Figure 1 accordingly. |

https://doi.org/10.5194/soil-2021-113-CC1

**RC2: Anonymous Referee #2, 11 Jan 2022**

| *RC1 comments* | *Authors' answers* |
|---|---|
| In this study, the authors have evaluated how the conversion of conventional agriculture to conservation agriculture could affect soil physical properties. For this purpose, the authors have monitored different soil physical properties during 3 years in plots with different tillage treatments and different cover crops. These soil physical properties were the bulk density (BD), penetration resistance (PR), hydraulic conductivity (Ks), and sorptivity (S). The work results showed that the absence of tillage enhances soil physical properties. At the same time, the use of some cover crops also improves the soil physics. In general, the research makes sense since it looks to increase the knowledge about the effects caused in the soil during the transition to conservative agriculture. However, the manuscript needs a few improvements before its publication. Some parts of the text are a little difficult to read. The experimental method could be clarified to improve its understanding. Moreover, in the results section, there is too much information in parentheses. I would recommend only writing the necessary numeric values to well describe the work results. Some parts of the text should be rewritten to do it more readable and intelligible. Finally, the part of references shows some little mistakes. I specify them below. Please, correct them. | We thank the reviewer for the precious comments. We improved the manuscript accordingly. Particularly, we better described the methods as observed by all the reviewers, we also clarified the results, highlighting the significant difference while summarizing the other information. The references was carefully revised to avoid inhomogeneity. Finally, the whole manuscript was revised by a professional English reviewer to guaranty the language correctness and clarity. |
| L10. I would recommend to write the short version of bulk density and penetration resistance in parenthesis the first time that appear in the text. | We modified accordingly, see e.g., L10, 11 and 13. |
| L10. I consider that 'soil hydraulic measures' is unspecific. I would recommend to be more specific when writing an abstract. Please, change this to 'saturated hydraulic conductivity (Ks) and sorptivity (S)'. | We added this information in L11. |
| L25-28. Please, add some references that support it. | We added this reference: HOBBS, Peter R.; SAYRE, Ken; GUPTA, Raj. The role of conservation agriculture in sustainable agriculture. Philosophical Transactions of the Royal Society B: Biological Sciences, 2008, 363.1491: 543-555 in LL31-32. |

| | |
|---|---|
| L74-77. There was no hypothesis in the sentences where you defined the aims of the research. What were the expectations for this research? What results did you expect to obtain? On what previous evidences were based your expectations? | We clarified the starting hypothesis in LL119-121. |
| L95. BD and PR have been already used before (L66-77). Write in parentheses only the first time you mentioned. | We modified the text accordingly. |
| Figure 1. Why were not bulk density and penetration resistance analysed in 2019? | Both BD and PR are invasive tests and excessive repetitions could impact on soil structure. BD particularly requires heavy machinery which could cause soil compaction, while penetration resistance was performed with many replicates, which results in soil disturbance. Thus, we retained more important to have two measures in the last experimental year, to monitor the evolution of these parameters along a single growing season, when the first effects of conversion to CA a should start to be evident. In fact, literature reports long conversion time and often in the first years negative and positive CA effect are not easily assessable. |
| L114-115. Why was not the penetration resistance analysed in 2018 (time 0)? Please, explain it. | As mentioned in the first comment, all the sampling we performed are destructive, and required specific pedoclimatic conditions together with field accessibility, and absence of the main culture. Particularly PR required enough soil moisture, and the studied soil results often too dry for this analysis. |
| L128-129. Were the normality and homoscedasticity of data checked? Please, specify it. | We tested these properties. We added this information in LL186-187. |
| L139-140. Define GWC in the Table 1 caption. | We revised accordingly. |
| L165-168. It would be interesting to know if there were differences in the penetration resistance among the different cover crop for each tillage treatments every 10 centimetres along the soil profile. Were these differences analysed? If affirmative, were significant these differences? | We did not mention the CC*Depth results since it was not involved in the model according to the lowest AIC criterion. |
| L325. The reference is not correct. The name of authors and the year of publication are missing. Please, correct it. | We revised this reference (L441). |
| L356. The DOI appears twice. Please, correct it. | We removed the repetition (L476). |
| L363. The DOI is missing. Please, correct it. | It is a book chapter and the DOI does not appear according to Journal citation format. |

| | |
|---|---|
| L406. See comment for line L363. | It is a book chapter and the DOI does not appear according to Journal citation format. |
| L429. See comment for line L363. | It is a book chapter and the DOI does not appear according to Journal citation format. |

https://doi.org/10.5194/soil-2021-113-RC2